# “Show Me What You Got”: The Nomological Network of the Ability to Pose Facial Emotion Expressions [note 1]

**DOI:** 10.3390/jintelligence12030027

**Published:** 2024-02-26

**Authors:** Mattis Geiger, Sally Gayle Olderbak, Oliver Wilhelm

**Affiliations:** 1Health Communication, Implementation Research, Bernhard Nocht Institute for Tropical Medicine, Bernhard-Nocht-Straße 74, 20359 Hamburg, Germany; 2Health Communication, Institute for Planetary Health Behaviour, University of Erfurt, Nordhäuser Str. 63, 99089 Erfurt, Germany; 3Epidemiological Survey of Substance Abuse, Mental Health and Addiction Research, Institut für Therapieforschung (IFT), Leopoldstr. 175, 80804 Munich, Germany; olderbak@ift.de; 4Department of Individual Differences and Psychological Assessment, Institute for Psychology and Education, Ulm University, Albert-Einstein-Allee 47, 89081 Ulm, Germany; oliver.wilhelm@uni-ulm.de

**Keywords:** emotion expression, posing, socio-emotional abilities, nomological network, measurement models

## Abstract

Just as receptive emotional abilities, productive emotional abilities are essential for social communication. Although individual differences in receptive emotional abilities, such as perceiving and recognizing emotions, are well-investigated, individual differences in productive emotional abilities, such as the ability to express emotions in the face, are largely neglected. Consequently, little is known about how emotion expression abilities fit in a nomological network of related abilities and typical behavior. We developed a multitask battery for measuring the ability to pose emotional expressions scored with facial expression recognition software. With three multivariate studies (*n*_1_ = 237; *n*_2_ = 141; *n*_3_ = 123), we test competing measurement models of emotion posing and relate this construct with other socio-emotional traits and cognitive abilities. We replicate the measurement model that includes a general factor of emotion posing, a nested task-specific factor, and emotion-specific factors. The emotion-posing ability factor is moderately to strongly related to receptive socio-emotional abilities, weakly related to general cognitive abilities, and weakly related to extraversion. This is strong evidence that emotion posing is a cognitive interpersonal ability. This new understanding of abilities in emotion communication opens a gateway for studying individual differences in social interaction.

## 1. Introduction

If, in fact, *homo est animal rationale sociale*[note 1] (“humans are rational and social animals”), then it is no surprise why emotion is among the most researched topics in psychology. Emotions are expressed in humans and other animals ([17]; [19]). They interfere with rational choices but facilitate decision making when resources are short ([52]), and they have a communicative function crucial for any social interaction ([95]). In this paper, we investigate individual differences in facial emotion expression ability, a crucial aspect in socio-emotional communication, and embed this ability in a nomological network of personality, intelligence, and socio-emotional abilities.

### 1.1. Individual Differences in Emotion Communication

The component process definition of emotion ([95]) defines five components of emotions (appraisal, bodily symptoms, motivation, motor expression, and feeling) and links them with specific functions. The motor expression component is crucial for human communication. It triggers and drives the somatic nervous system to exhibit automatic communication of an emotional state ([95]); i.e., a person in fear might automatically express a fearful facial expression to warn others or to trigger them to help. The idea of emotions serving a communicative function is proposed by most theories of emotion. Throughout this paper, we will focus on the facial domain of emotion communication, because this is the only expressive domain in which objective systems to evaluate facial expressions currently exist ([33]).

Emotions as a fundamental form of communication are far from flawless. For example, in movies, we see actors expressing emotions at their best and, only because of the context, do we expect that the expressed emotion was just acted. We typically have a concept of good and bad emotion posing and will judge that actor’s performance accordingly. In other words, we assume individual differences in the ability to express emotions. In emotion perception and recognition research—the receiving end of this communicative path ([96])—individual differences are well-documented ([44]; [99], [100]; [119]). Hence, the idea of individual differences in the ability to express emotions is plausible and has been proposed earlier (e.g., [103]). Given that we expect individual differences in the ability to express emotions, it is a promising ability to examine when striving to extend models of socio-emotional abilities. Furthermore, given that perceptive socio-emotional abilities have been embedded in models of intelligence, too ([43], [44]; [66]; [76]; [101]), emotion expression abilities might also be a promising addition to models of intelligence.

### 1.2. Earlier Approaches to Emotion Expression Ability

The idea to study individual differences in emotion expression ability is not new ([14]; [38]; [61]; [88]). Yet, existing measures of emotion expression ability do not adhere to standards of aptitude testing ([27]). Consequently, our understanding of this construct is insufficient. Criteria for maximal effort tests include maximal performance instructions, an understanding or reasonable assumption of test-takers emotional states, multiple independent items/multiple tests that allow for latent factors models, a standardized assessment of emotion expression behavior, an evaluation of behavior with regards to veridicality, and scores that allow one to capture individual differences, i.e., continuous scores ([15]). 

Most emotion expression research uses self-report questionnaires in which participants respond to items such as “I think of myself as emotionally expressive” ([55]) or “I’m usually able to express my emotions when I want to” ([82]). Self-report questionnaires mostly assess typical behavior and not maximal performance and the few self-report measures intended to measure the same ability as a maximal performance construct hardly correlate ([48]; [80]; [81]). Thus, we do not consider these approaches viable for measuring emotion expression ability. 

Another established approach in emotion expression research is the expressive accuracy approach (e.g., [89]; [88]), in which participants are asked to express certain emotions that are then evaluated by a group of raters. Similarly, in the slide-viewing technique, individuals are instructed to express emotions based on colored slides and then their emotional expressions are rated by another group of judges ([8]). If these groups of raters were sufficiently large and a representative sample of the general population, they might qualify as a viable option for measuring emotion expression abilities. However, usually, rater samples are small and not representative, thus largely diminishing the objectivity of this approach.

Finally, there are approaches in which participants experience a form of emotion induction and their expression behavior is judged by more objective means such as the Facial Action Coding System ([26]). For example, they startle participants or emotionally trigger them with videos while instructing them to regulate their facial expressions. Then, participants’ facial responses are evaluated via objective facial action coding ([14]). Although these approaches fulfill most criteria of aptitude testing, they do not fulfill all. Even the established methods of emotion induction used in studies with this approach, such as pictures (e.g., [60]) or videos (e.g., [91]), do not permit individual standardization of the intensity of the induced emotion. Thus, the test taker’s real emotional state is unknown. Furthermore, prior work with this approach was not multivariate enough to allow for estimating latent factors. And, lastly, human raters, even Facial Action Coding-trained ones, are not as objective in their rating as computer software would be ([33]).

We conclude that none of the prior approaches to assess emotion expression ability fulfill all criteria of aptitude testing. Consequently, any understanding of the constructs, such as how they relate to other socio-emotional abilities, how they can help to enhance models of intelligence ([28]; [27]; [66]), or how emotion expression ability among other socio-emotional abilities relates to personality, is still limited. 

In this paper, we strive to fill this gap. We follow three steps. First, we develop multiple emotion expression ability tests that adhere to all criteria of aptitude testing. Second, we define competing measurement models based on attributes of the constructs reflected in the items of our tests and test them in empirical studies. Third, we use these tests and embed them in a nomological network to learn more about overarching models of socio-emotional abilities and traits and about models of intelligence. 

### 1.3. Step I: Objective Emotion Expression Measurement 

To develop new maximal effort emotion expression tasks, we focused on the two criteria that not even the most advanced approaches to measuring emotion expression match: (1) we established an understanding of the emotional state of the test taker and (2) we developed a standardized system involving computer software to record and objectively evaluate facial expressions with regard to veridicality (see “Scoring Facial Expressions” below) that is time-efficient and allows for more multiple items and tests. We address the evaluation of expressions by extending prior research ([75]), which we describe below. 

To answer the first criterion, we first considered which types of emotion expression behavior exist and which qualify for tests of emotion expression ability. Emotion expression behavior can be categorized into three sets based on the situation from which they arise: (1) an emotion is felt, and a congruent emotion is expressed; (2) an emotion is felt, and an incongruent expression is produced; or (3) no emotion is felt, but emotion is expressed. All expression behaviors in these categories are either driven by push factors (automatic behavior), pull factors (controlled behavior) of emotion expression behavior (c.f. [96]), or an interaction of both factors.

In the first category, there are two types of expression: (1a) We show a genuine facial expression when we lack (cognitive) resources or reasons to regulate automatically triggered facial expression and simply let automatic emotion expression emerge and fade according to the underlying emotional state—behavior solely driven by push factors. (1b) We enhance an emotion expression to increase the intensity of our communicative signals, for example, to increase the probability of being correctly understood. When enhancing, the push factors of emotion expression are supported by pull factors.

Within the second category, there are also two types of expression: (2a) We suppress our emotional expression and instead present a “poker face” (i.e., a neutral facial expression) or reduced emotional expression (toward neutral) despite our emotional state triggering an emotional expression. When neutralizing, the push factor of emotion expression is inhibited. (2b) We mask how we feel by expressing another emotion. For example, we might be disgusted by a certain dish but in order to not insult the cook, we keep smiling. Thus, a pull factor overrides the push factor of emotion expression.

Finally, the third category contains two types of expression: (3a) Facial muscles are activated just by seeing a corresponding facial expression of another person; however, no emotion is felt ([21]; [59]), which is also called mimicry—a pure push-factor driven behavior. (3b) We pose an emotion, while initially feeling no emotion, to help communication (e.g., signaling empathy)—a pure pull-factor driven behavior. For example, someone frowns and clenches teeth in a painful expression to signal empathy toward a person who is actually in pain, without personally feeling pain. 

Genuine expression and mimicry are automatic behaviors and can be classified as typical behavior constructs, defined as behaviors individuals are likely to display in everyday behavior and on a regular basis (c.f., [15]). Enhancing, neutralizing, masking, and posing can be described as abilities, or maximal effort constructs, defined as behaviors exhibited when one is motivated to try to achieve their best in a given task ([15]) and, thus, define emotion expression abilities. 

Among these four, only posing requires a non-emotional state to begin with, which can be assumed to be the typical state of participants in laboratory studies (e.g., see the neutral group in [84] or reactions to neutral films in the laboratory in [83]). Obviously, without objective measures of emotional states, we cannot know the emotional state of participants in a laboratory study and, thus, the other three expression behaviors are not a viable option, either. Although it is also unclear whether all participants are equally unemotional in a lab setting—the requirement for posing—assuming a neutral state in lab studies presumably introduces less error than assuming emotional states after unstandardized emotion induction methods. Furthermore, although the process of posing itself might induce emotions, we assume these effects cancel each other out by asking participants to express sequences of different emotions with only short inter-item intervals. Consequently, we chose to assess individual differences in the ability to pose facial expressions of emotion as a first step toward measuring individual differences in emotion expression ability. For the sake of brevity, throughout this manuscript, the term expression ability will always refer to the ability to pose facial expressions. 

Posing facial expression items can differ by the target emotion to be posed (e.g., disgust) and the stimulus type instructing target emotions (e.g., emotion words or emotional faces). Emotions to be posed should be aligned with expression scoring systems. So, if a scoring system is based on the six basic emotions ([25]), tests should contain items for all six basic emotions. The standard in instructing which emotion is to be posed clearly is using emotion words, such as “happy” or “surprised”, and asking participants to produce expressions based on their individual understanding of the expression. However, in everyday life, we also imitate others’ expressions. For a posing facial emotion expression test, this means a test in which participants see facial emotion expressions and imitate them is also a viable option.

To answer the second criterion required for improved emotion expression ability tests, the standardized recording and time-efficient objective evaluation of facial expressions, it is vital to rely on recent technological advances. Although well-trained human raters (e.g., FACS [Facial Action Coding System; [26]]-trained) can achieve high interrater reliabilities (e.g., [124]), their ratings are less objective than perfectly replicable ratings from a machine, and machines achieve equal or higher accuracy scores compared with human raters ([57], [56]). Additionally, machine ratings are cheaper and quicker than human raters, allowing for analyses in higher time resolution and for more recordings in large samples ([33]). Therefore, we use state-of-the-art emotion recognition software to score the posed facial expressions of test takers. 

### 1.4. Step II: Measurement Model of Emotion Expression Ability 

To better understand emotion expression ability, we must demonstrate the construct’s validity by establishing a measurement model of emotion expression ability. Measurement models and respective confirmatory factor models in prior work exist for some self-report questionnaires, but they cannot be extended to emotion expression abilities for the reasons discussed earlier. 

As introduced in Step I, emotion expression ability tests can vary by the emotion to be displayed (six basic emotions) and the task type (word vs. image stimuli, i.e., production tests vs. imitation tests) used to instruct the target expression. Whether these distinctions result in specific factors or whether individual differences in emotion expression ability tests are best explained by a general factor is an empirical question that can be solved by comparing competing measurement models. That is, measurement models may include a general factor of emotion expression or correlated task or emotion factors. Models can also vary by having no specific factors or specific emotion and task factors. 

A general factor would represent the broad ability to express emotions. Task-specific factors and emotion-specific factors would represent either method variation or variation due to specific abilities. For example, the computer software and scoring algorithm may systematically contribute to stable individual differences in performance (e.g., software-derived emotion scores differ in range; see [10], [9]) or, because posing different emotions to a camera might be perceived as differently unusual, common variation may arise. By comparing models with regard to fit and parsimony, we can choose the best model to continue with research questions about the factor(s) of emotion expression ability.

### 1.5. Step III: Nomological Network of Emotion Expression Ability

After concluding on a measurement model of emotion expression ability, we can test how this ability relates to other socio-emotional abilities, intelligence, and extraversion. We can embed emotion expression ability in a nomological network of convergent and divergent constructs ([16]) to test whether this construct is a viable addition to models of socio-emotion traits and intelligence. Below, we present a hypothesized nomological network of why and how emotion expression ability should relate to established constructs. The constructs are sorted by nomological proximity to emotion expression ability.

#### 1.5.1. Non-Emotional Expression

All forms of facial expression, emotional or not, serve a communicative purpose. Similarly, all facial expressions are controlled via the volitional circuit of the *n. facialis* (for a comparison of volitional and emotional circuits, see [46]; [109]). Thus, the ability to express non-emotional expressions (e.g., lowering mouth corners, details see below) should be strongly related to the ability to express emotions. From a psychometric perspective, both emotional and non-emotional expression tasks can share the same type of productive responding, the same scoring approach, and consequently, any possible artifacts due to the specificities of expression coding methodology. Nevertheless, they differ by specific (emotional vs. non-emotional) context, so their relation can be expected to be strong, but not unity.

The overlap of emotional and non-emotional expression abilities is similar to the overlap of facial identity and emotion perception and recognition. Facial identity perception and recognition refers to the common variation in individual differences in perceiving, learning, and recognizing information about facial identity. This construct was demonstrated to be a specific ability distinct from general cognitive ability (e.g., [43]). Similarly, facial emotion perception and recognition refers to the individual difference ability to perceive, learn, and recognize emotion in unfamiliar faces. Both facial identity and emotion perception and recognition correlate strongly ([44]). This is no surprise given that both abilities rely on the same neural circuits of facial information processing ([40]; [39]), require the processing of the structural code of facial information ([6]), and are receptive, basic socio-emotional abilities ([44]). Furthermore, the tasks for both constructs share the same basic methodology (i.e., nature of stimuli and their presentation, type of response behavior, and performance appraisal). Consequently, we expect a similarly high correlation between non-emotional and emotional expression abilities as reported for facial identity and emotion perception and recognition. 

##### Receptive Socio-Emotional Abilities 

Theories of language acquisition (e.g., [64]) propose that productive and receptive abilities develop together over the course of the lifespan. This is partially supported by socio-emotional abilities, where researchers found a small correlation (*r* = .19) between tests of expressing and perceiving non-verbal information in studies with communicative intent ([28]). This positive correlation is also no surprise given that producing and perceiving emotion expressions are fundamental in communication. As abilities representing receptive socio-emotional abilities, we selected prominently studied socio-emotional constructs including facial identity and emotion perception and recognition, recognition of emotional postures, emotion management, emotion understanding, and faking ability. 

We chose facial emotion and identity perception and recognition because they focus on the facial domain and thus share the same expressive channel with our emotion expression assessment. Correlations between emotion perception and emotion expression were stronger between tasks from facial or body channels compared with the vocal channel ([28]). Similarly, tests of emotion recognition with different channels correlated weaker ([99]) relative to tests with the same channel ([44]). Thus, we expect medium to strong correlations between tests of facial emotion expression and facial emotion perception and recognition, and slightly lower correlations with facial identity perception and recognition because the latter do not share the emotional component. Similarly, we expect lower correlations, i.e., of medium effect size only, with emotional posture recognition because the latter does not share the same expressive channel. 

Following models of emotional intelligence, emotion understanding and management should be positively related to emotion expression abilities (e.g., [27]). Both abilities involve handling emotional expressions (by understanding or managing them) but they are also heavily situationally dependent ([69]), whereas emotion expression abilities can be assessed as situationally independent. Emotion understanding and management would best be assessed with behavioral observation in different emotional situations. However, because this is hardly achievable while still adhering to standards of psychometric testing, in research, emotion understanding and management are typically assessed with situational judgment tests (SJTs). SJTs of emotion understanding and management resemble receptive ability tests, but oftentimes their veridicality is questionable, thus not fully qualifying as ability tests ([117]). Furthermore, due to their test design that involves reading complex vignettes and response options, SJTs involve more verbal literacy than any of the previously mentioned tests, and they do not solely focus on communicating via the facial channel. In sum, although conceptually closely related to emotion expression, emotion management and understanding are assessed with limited measurement approaches. Consequently, we expect them to only have small correlations with emotion expression ability. 

With its communicative function, emotion expression also plays a role in deception. A good lie means aligning content and behavior, including emotional expressions ([114]), and the capacity to express non-felt emotions should be positively related to deception skills. We chose to focus on faking ability, a socio-emotional ability based on the ATIC (Ability To Identify Criteria) model ([54]). It represents the ability to identify what psychological tests or interviewers in assessment situations “ask for” or “want to hear” and successfully respond accordingly. Faking ability correlates substantially with facial emotion perception and recognition and general cognition ([35]). We studied faking ability as the ability to fake a desired personality profile on a questionnaire. Given that tasks share neither the facial nor the emotional component, we expect a small correlation between faking ability and emotion expression.

##### Non-Socio-Emotional Abilities

The most consensual model of individual differences in intelligence, the Cattell–Horn–Carroll model (CHC model; [71]), is based on a positive manifold among tests of cognitive abilities. In other words, tests of cognitive abilities all correlate positively, implying a general factor of intelligence ([104]). Recent work with ability tests of emotional intelligence demonstrated that this positive manifold might be expanded to emotional abilities and that these can be included as stratum I or II abilities/factors in the CHC model ([66]). In this study, the authors only used the Mayer–Salovey–Caruso Test of Emotional Intelligence ([70]) to measure socio-emotional abilities, so their study lacked tests of emotion expression. Currently, no systematic investigation of relations between emotion expression and the general factor of intelligence *g* exists. However, as emotion expression has been demonstrated to correlate with other emotional abilities ([28]) and, thus, can be assumed at stratum I under the stratum II factor of socio-emotional abilities, we hypothesize that emotion expression abilities are subject to a positive manifold and thus relate positively to *g*. In our studies, *g* will be indicated by measures of fluid intelligence, working memory capacity, and immediate and delayed memory. From a psychometric perspective, these tests only share the attribute of a maximal performance test with emotion expression. Consequently, we expect the correlation between emotion expression and the g factor and its underlying abilities to be weak. 

Following prior work correlating general cognitive abilities and emotional abilities ([76]), we differentiate crystallized intelligence (accumulated skills and knowledge, and their use) from general cognitive abilities (*g*). For successfully posed emotion expressions, knowledge about emotions and their typical expressions is required. Although such emotional knowledge might be considered highly specific, recent research on the dimensionality of factual knowledge demonstrated how closely even the most diverse knowledge domains are related ([106]). Consequently, we expect small correlations between emotion expression and general knowledge, a marker variable to crystallized intelligence.

##### Typical Behavior

Finally, assuming that higher levels of socio-emotional personality traits, such as extraversion, result in more social interaction and therefore more situations in which one needs to communicate one’s emotional states, we assume a weak correlation between emotion expression and self-reported extraversion. We only expect a weak correlation because (a) self-report is prone to response biases that presumably distort validity and (b) extraversion is a typical behavior construct, which is usually weakly related or unrelated to cognitive abilities ([79]; [117]). 

Similarly, it could be argued that self-report measures of mixed model emotional intelligence (mixed model EI) such as the Trait Emotional Intelligence Questionnaire (TEIQ; [32]) and the Trait Meta Mood Scale (TMMS; [93]), which assess socio-emotional personality traits, too, should correlate with emotion expression ability. In prior work, they have already been shown to correlate weakly with receptive emotional abilities (e.g., [18]; [92]; [123]) and, thus, a similar relation with emotion expression ability could be expected. However, given that measures of mixed model emotional intelligence have been shown to be largely overlapping with the Big Five, including major correlations with extraversion ([111], [112]), any relations between mixed model EI and socio-emotional abilities might vanish when variance in mixed model EI is controlled for by the underlying socio-emotional personality trait extraversion. In fact, two of our studies included measures of mixed model EI, which allowed us to test this idea. These analyses are reported in the Appendix A. 

### 1.6. Current Study

The purpose of this paper is to show that facial emotion expression abilities can be measured according to established psychometric standards and that the construct measured is a valuable addition to research on socio-emotional traits and intelligence by testing correlations in a nomological network of constructs. We try to accomplish these goals in three steps: 

I. Task development: Before reporting the results from our three consecutive studies, we introduce our expression tasks that adhere to standards of maximal performance testing. With experimental control of the presentation of task instructions and items, along with the computerized recording, coding, and scoring of responses, we maximize the objectivity of our measurement. Although the tasks we introduce have been used in other already published studies as covariate constructs ([34]; [78]), this manuscript is the first to extensively describe the construction rationale of the test, as well as construct validation efforts with regard to the measurement model and the position of emotion expression abilities in a nomological network of related abilities and typical behavior traits.

II. Psychometric evaluation: To evaluate factorial validity, we test whether individual differences in the indicators derived from our newly developed test are accounted for by sound measurement models ([5]). A sound measurement model is a confirmatory factor model designed in accordance with theoretical considerations to explain individual differences in indicators of a psychological test. They offer the opportunity to test whether the assumed factorial structure of a test matches with empirical reality. For our emotion expression ability tests, we establish a general ability to pose emotional expressions as a latent variable and test competing measurement models in three studies. 

III. The nomological network: We validate the general ability to pose emotional expressions by investigating its position in the nomological network of socio-emotional and general cognitive abilities, as well as self-reported socio-emotional traits. Across our three studies, we test correlations between the general ability to pose emotional expressions and non-emotional expression ability, receptive socio-emotional abilities (perceiving and recognizing identity and emotion in unfamiliar faces, emotion management, emotion understanding, emotional posture recognition, faking ability), general cognitive abilities, and the personality factor extraversion (see Table 1 for a summary). Following the critique of frequentist significance testing ([11]; [53]), we report both effect sizes and *p*-values, but we focus on effect sizes when interpreting the results. Because effects that replicate would become significant if tested in bigger samples, instead of discussing significance, we focus on the consistency and replicability of our results across the three studies.

### 1.7. Step I: Facial Expression Ability Task Development

#### 1.7.1. Task Design

First, we developed two emotion expression tasks in which participants were instructed to pose emotional expressions with their faces to the best of their ability while being videotaped. In the first task, called production, participants received a written prompt naming the target that should be expressed, i.e., one of six basic emotion words (anger, disgust, fear, happiness, sadness, or surprise), and were asked to produce the respective facial expression. In the second task, called imitation, pictures of faces expressing target expressions were presented, and participants were asked to imitate the target face.

The trial presentation followed a uniform schedule. Participants had 10 (study 1) or 7 s (studies 2 and 3) to read the written prompt (production) or view the target expression (imitation) and prepare their expression (preparation time). Then, participants had 5 (study 1) or 3 s (studies 2 and 3) to produce or imitate the intended emotion at their best capability (expression time). Expression time immediately followed preparation time and trials followed each other with an inter-trial interval of 200 ms. Preparation time and expression time were shortened for studies 2 and 3 after many participants in study 1 reported the preparation and expression phases felt somewhat lengthy. 

For the target emotions in the production and imitation tasks, we chose the six basic emotions ([25]). We additionally included neutral trials for baseline expression recordings (see below for details). As shown in Table 2, a timer in the top center of the screen, just below the camera, gave participants temporal orientation. The preparation phase instructions (emotion label or emotional face) were presented in the center of the screen. In the production task, we repeated each target once to increase task reliability. In the imitation task, target expressions were presented four times, but each trial included a new identity, and each emotion was expressed by two females and two males. Items were presented in fixed pre-randomized orders. Faces for the imitation task were sampled from previously unused (grey-scaled and ellipsed) faces of the BeEmo stimulus database that was also used to develop a task battery to measure facial emotion recognition and memory ([119]). Exemplary pictures of all six basic emotion expressions are presented in Figure 1A. We created two variants of the imitation task, namely, with and without feedback. In the feedback condition, participants saw their face next to the target face via a mirror, whereas there was no mirror in the without-feedback condition. Example items are shown in Table 2.

To evaluate the nomological network, we additionally developed one non-emotional production task as a closely related covariate measure, which was always presented as the first expression task. The goal of this task was to assess the ability to pose facial expressions independent of emotional expressions. To sample expressive abilities in the whole face, keep the testing time short, and create items that were not too difficult, we selected two distinctly executable facial movements per major facial region (eyes, nose, mouth; for a compendium of facial movements, see [26]) as targets. Participants were instructed to display their facial movements as best as possible. Around the eyes, we asked participants to raise their eyebrows (AU1 and AU2; originate from *m. frontalis*) or to furrow them (AU4; originates from *m. corrugator*). Around the nose, we asked participants to widen their nostrils (AU38; originates from *m. nasalis*) or wrinkle their nose (AU9 and AU10; originate from *m. levator*). Around the mouth, we asked participants to move the corners of their mouth as high (AU12; originates from *m. zygomaticus*) or as low (AU15; originates from *m. depressor*) as possible. The position and line of movement of these AUs are shown in Figure 1B, and an example item is presented in Table 2. Each trial was presented twice. Additionally, for the very first item, participants were asked to present a neutral, relaxed expression (i.e., do nothing), so they could get used to the timed task design.

The production expression tasks are available on the project’s OSF page (https://osf.io/9kfnu/ (accessed on 2 February 2024)). Due to privacy restrictions of the facial stimuli, the imitation task is only available upon request from the corresponding author and after signing a user agreement. All other measurement instruments used in the studies are available from the authors upon request. 

#### 1.7.2. Scoring Facial Expressions

A defining characteristic of ability tasks is that they need a veridical response for every item ([15]). For the non-emotional production task, the veridical response was the maximal value that could be achieved for the AUs underlying the requested facial movement. Defining a veridical facial emotion expression is more difficult, but the communicative function of emotion ([95]) offers a solution: communication can be successful or not. Research on the universality of emotion by [25] ([25]) demonstrated that what he calls universal or prototypical expressions are those expressions understood globally. We can conclude that the more prototypical an emotion expression, the higher the probability that it is identified correctly by another person, thereby maximizing its communicative success. Given that communication is maximally stressed in posing expressions, we argue that maximizing the communicative success of expressions is the goal of an expression task and, therefore, a maximally prototypical expression is the veridical response to an item of such a task.

Scoring both facial movements and prototypical emotional expression can be achieved with standardized facial expression coding systems, such as the Facial Action Coding System (FACS; [26]). Emotional expressions are transient and dynamic phenomena and should therefore be recorded on video for thorough evaluation. Throughout all studies, we recorded high-definition videos with a framerate of 25 frames per second (fps), producing several thousand frames per participant. To satisfy the dynamic nature, every single frame had to be rated. Human raters, as typically used in FACS, are neither efficient nor objective enough to evaluate millions of frames of facial expressions. However, FACS has inspired the development of numerous facial expression recognition software programs (for a review, see [12]) that solve the problems of human ratings and have been shown to be valid and reliable tools for research ([24]; [33]; [57]; [58]). 

Typically, such software is capable of scoring both some emotion scores—basic emotions and/or positive and negative valence—and a set of AUs. We use Facet with the Emotient SDK 4.1 ([29]), a tool with the reportedly highest accuracy rates in scoring emotion expressions and AUs of different software tools ([24]; [30]; please note that Facet cannot score AU38 activity, which is why this item from the non-emotional expression task was excluded in our analyses). We only score behavior during the expression time of every item and, following our earlier argument on veridical responses to expression items, we only consider the emotion score of the item target emotion. For example, if the target emotion of an item with a 5 s expression time is an angry facial expression in an imitation task and the video is recorded at 25 fps, we extract the respective 5 × 25 = 125 anger values in a time series, similar to data from physiological responses. We then loess-smooth the data (quadratic degree and a smoothing parameter α = .22; Olderbak et al. 2014), extract the maximum scores, and regress them on their respective baseline emotion expressions extracted from neutral expression trials to continue with the residual value. More details regarding this scoring approach are reported in [75] ([75]).

This process results in one ability score per trial indicating how well participants pose the target emotion. We analyze Facet evidence scores, which are log-transformed odds ratios with higher values indicating stronger target expression. Thus, the higher a participant’s score, the better their performance. Facet produces missing data when, for example, participants move their heads at too extreme angles (yaw, pitch, or roll deviation of |45|° or more; [30]) and, therefore, the software’s face recognition fails. Participants were instructed to always face the camera and remove obstructions to their faces. Thus, other things being equal, the proportion of missing data in Facet indicates the proportion of non-adherence to instructions. Therefore, we set a person’s item scores to missing if their respective time series had more than 20% missing data points.

#### 1.7.3. Data Processing and Analyses

After evaluating the expression videos with Facet, the output data processing (smoothing, max-score extraction, baseline treatment, missingness treatment) was conducted with SAS (version 9.4; [94]). To facilitate open science, we also wrote functions for all these steps in R and uploaded them to the OSF (https://osf.io/9kfnu/ (accessed on 2 February 2024)). All consecutive data processing and analyses were conducted in R (version 3.6.2; [85]), with the packages psych (version 1.9.12.; [87]), lavaan (version 0.6-5; [90]), and semTools (version 0.5-3; [50]). 

#### 1.7.4. Summary Step I

We develop two tasks that meet our requirements for developing a maximal effort test of emotion expression ability: (1) we can reasonably assume what test takers feel during the test, and (2) expressive behavior is evaluated in a standardized, efficient, and objective way and scored with regard to a veridical response. That is, we chose to focus on emotion posing only because the initial emotional state (neutral) of this behavior is assumed to be the default state during lab experiments. Next, we introduce measurement models of these tasks before we present the empirical studies in which the measurement models are tested and the nomological networks for Step III are analyzed. 

### 1.8. Step II: The Measurement Models of Emotion Expression Ability

To address Step II, identifying a measurement model of emotion expression ability, we first defined a set of six plausible measurement models (M1 to M6; see schematic depictions in Figure 2). They were designed to reflect and test three major sources of between-subject variation in emotion expression: (1) general emotion expression ability, modeled as one factor loading on all items, (2) task-specific variance, modeled with factors indicated by production or imitation trials, and (3) emotion-specific variance, modeled with six factors indicated by the relevant emotion trails or as correlations between the residuals of same emotion trials. We also tested variations in the models depicted in Figure 2, though none offered a better fit to the data. The data and analysis syntax for these additional model candidates are available in the Appendix A.

M1 models a single general emotion expression ability factor, M2 has two correlated task-specific emotion expression ability factors, and M3 has six correlated emotion-specific factors. In models M4 to M6, we combine these sources of variation in bifactor models to separate general and specific variance. M4 combines the general factor with the six orthogonal emotion-specific factors. M5 combines the two correlated task-specific factors with the six orthogonal emotion-specific factors. M6 combines all three sources of variation with a general factor, an orthogonal imitation factor, and six orthogonal emotion-specific factors. 

We tested M1 to M6 on item-level and parcel-level data, but for parsimony, the item-level analyses are only reported in the Appendix A. Parcels were preferred to reduce the complexity of our models ([63]). All parcels were calculated as mean values across sets of items. For the emotion (and non-emotional) expression tasks, parcels were estimated from trials that had similar item types. For example, all anger imitation items were combined to an average anger imitation score (or all AU9/10 [*m. levator*] items in the non-emotional expression task were combined to an average *levator* score). This resulted in 12 emotional production parcels, 12 emotional imitation parcels, and five non-emotional production parcels. 

All factors were identified with effects coding ([62]; please note that whenever a factor was based on two indicators only, respective loadings were set to equality for local identification) and correlations were tested with a likelihood ratio test ([37]) and an adjusted χ^2^-distribution ([108]). Model fit was deemed acceptable with CFI and TLI ≥ .90, RMSEA < .08, and SRMR < .11; the fit was deemed good with CFI and TLI ≥ .95, RMSEA < .05, and SRMR < .08 ([1]; [47]; [107]). Missing data were handled with full information maximum likelihood imputation.

## 2. Study 1

### 2.1. Methods

#### 2.1.1. Sample

We recruited 273 healthy adults between the ages of 18 and 35 living in the Berlin area. Due to technical problems and/or dropouts between testing sessions, 36 participants were removed because they either had: (1) no data for the emotion expression tasks or (2) missing data for more than five covariate tasks. The final sample of n = 237 participants (51.1% females) had a mean age of 26.24 years (*SD* = 6.15) and a reasonably heterogeneous educational background: 20.7% without a high school degree, including degrees that qualify for occupational education; 46.4% with a high school degree; and 32.9% with academic degrees. Within the largest SEM in this study, including 21 indicators, the sample size is sufficient to fulfill the five observations per indicator rule of thumb ([2]). With this sample, a significance threshold α = .05, and a moderate power of 1 − β = .80, we have enough sensitivity to detect bivariate manifest correlations as small as r = |.16| (calculated with G*Power; [31]).

Data from this sample was used in earlier publications on facial emotion perception and recognition ([44]; [119]) and expression scoring ([75]). However, these data have not been used to study measurement models of emotion expression ability or the nomological network of this ability. 

#### 2.1.2. Procedure, Constructs, and Measures

Data were collected in three test sessions, each about three hours in length (including two breaks), using Inquisit 3.2 ([23]), and participants were tested with five to seven days in between sessions. We will only describe measures that are relevant to our research questions. The presentation of measures is organized around the construct they assess. In terms of the order of tasks, emotional and non-emotional tasks were alternated, if possible. For practical reasons, i.e., the camera recording, the expression tasks were presented consecutively. Facial emotional stimuli used across different tasks never had an identity overlap across tasks. This means no two tasks with such stimuli contain stimuli from the same person. The tasks were selected because of their strong psychometric properties demonstrated in previous work, which we briefly highlight below. We report the reliability of the constructs using the factor saturation estimate ω, along with their measurement model, in the Results Section. For the sake of brevity, we will not include detailed task descriptions when constructs are indicated by more than two different tasks, for example, the construct facial emotion perception and recognition. 

Facial Emotion Expression Ability. We used two tasks to measure emotion expression abilities. For imitation, we tried two variants, one with and the other without feedback (via a mirror). Although the two imitation variants were designed to have 24 trials each, due to a programming error, only the first 12 trials in each task were presented. Therefore, both imitation tasks had 24 trials in total that were unevenly distributed across emotions: four anger, four disgust, four fear, two happiness, five surprise, and five sadness trials. 

Non-Emotional Facial Expression Ability. The non-emotional expression task (as described in Step I) was presented twice, before and after the facial emotion expression ability tasks. The trial order was pre-randomized within tasks. 

Facial Emotion Perception and Recognition (FEPR). Participants completed seven tasks from the BeEmo test battery ([119]): three perception tasks (“identification of emotion expressions from composite faces”, “identification of emotion expressions of different intensities from upright and inverted dynamic face stimuli”, and “visual search for faces with corresponding emotion expressions of different intensities”) and four recognition tasks (“learning and recognition of emotion expressions of different intensities”, “learning and recognition of emotional expressions from different viewpoints”, “cued emotional expressions span”, “memory game for facial expressions of emotions”). All tasks are reliable (ω = .59–.87; [119]) and valid (see e.g., [44]). The tasks were scored according to recommendations from the original authors, preferring unbiased hit rates ([115]) when recommended.

Facial Identity Perception and Recognition (FIPR). This construct was measured with six tasks from the BeFat test battery ([42]): three perception tasks (“facial resemblance”, “sequential matching of part-whole faces”, and “simultaneous matching of spatially manipulated faces”) and three recognition tasks (“acquisition curve”, “decay rate of learned faces”, and “eyewitness testimony”). All tasks are reliable (ω = .54–.90; [42]) and valid (see e.g., [43]). They were scored according to recommendations from the original authors.

Posture Emotion Recognition (PER). This ability was measured with the Diagnostic Analysis of Nonverbal Accuracy 2 (DANVA-2) posture task. Participants saw a picture of a person’s body with a blacked-out face who was making emotional postures and selected which among four response options (angry, fearful, happy, sad) best described the bodily expression. The task consists of 24 items and is reliable (α = .68–.78) and valid ([73]).

Emotion Management (EM). To assess this ability, we used a situational judgment test (Situational Judgment Test of Emotion Management; STEM). In this task, participants completed multiple-choice questions and selected the best reaction to an emotional situation. The task is reliable with α = .61–.72 ([65]). We used a short version (20 items) translated into German ([45]).

Emotion Understanding (EU). This ability was assessed with the situational judgment test of emotion understanding (STEU). In this task, participants select which among four options best describes how they or others would feel in an emotional situation. The task is reliable with α = .43–.71 ([65]). We used a short version (25 items) translated into German ([45]).

General Mental Ability (*g*). As indicators of *g*, we assessed fluid intelligence, working memory capacity, and immediate and delayed memory. Fluid intelligence was assessed with 16 items of the Raven’s advanced progressive matrices task ([86]). In this figural task, participants use reasoning to fill the lower right cell of a 3 × 3 matrix containing symbols. The task is reliable with a retest reliability of r_tt_ = .76–.91 ([86]).

We used well-established binding (number-position) and complex span (rotation span) tasks to assess participants’ working memory capacity, programmed according to [118] ([118]). The rotation span (ω = .84) and binding task have good reliability (ω = .80; [118]).

Memory without facial stimuli was assessed using six tests of immediate or delayed memory with either purely verbal, verbal–numerical, or visual stimuli (symbols). These tasks were adapted from the Wechsler Memory Scale ([41]; for details see [43], [44]; [120]).

Extraversion. We assessed extraversion (E) using the respective subscales from the NEO-PI-R ([13]). The extraversion scale of the NEO-PI-R has 48 items from six facet scales each with eight items, and it is reliable with α = .89 ([13]).

### 2.2. Results

#### 2.2.1. Step II: Measurement Models of Facial Emotion Expression Ability

To address Step II, establishing a measurement model of emotion expression ability, the 12 indicators of emotion expression ability were modeled, as shown in Figure 2. Model fit for all six models is summarized in the left-hand part of Table 3. Only M6 reached acceptable fit levels across the four fit indices. Consequently, we determined that M6 was the best-fitting measurement model. M6 models all three earlier-introduced sources of variations (general ability factor, task-specific variation, and emotion-specific variation). Please note that due to the large number of factors relative to only 12 manifest indicators, when estimating M6 jointly with other constructs, the models often had estimation problems, such as non-positive definite Ψ-matrices. Because our focus lies on the general factor, and to avoid the issue of non-positive definite Ψ-matrices in model estimation, we also defined a model that is a structurally equivalent model to M6, in which emotion-specific variation is represented by correlated residuals of indicators of the same emotion instead of factors. This model differs by one df because when identifying a bifactor model without a reference factor via effects coding, the intercept constraint for the general factor is redundant with the intercept constraints of the bifactors. This structurally equivalent model, which we call M6b, also had acceptable to good model fit (χ^2^(42) = 91, *p* < .001; CFI = .947; TLI = .916; RMSEA = .070; *SRMR* = .050). The general factor (ω = .663) and the task-specific imitation (ω = .520) factors were reliable.

#### 2.2.2. Step III: Nomological Network

Before estimating correlations with other constructs, we first evaluated the factorial validity of the covariate constructs. Then, to understand correlations between emotion expression ability and each construct, we constructed separate models relating emotion expression ability with individual covariate constructs. Please note we additionally examined relations with the receptive ability covariates using a bifactor structure (c.f., [7]), which takes the overlap of covariate constructs into consideration. For the sake of parsimony, these results are only reported in the Appendix A. 

The measurement models of the covariate constructs used composite task scores as indicators (all models are depicted in the Appendix A). We clustered the nomological network results in three steps: 1. emotion expression ability with non-emotional expression ability; 2. emotion expression ability with receptive performance measures; and 3. emotion expression ability with extraversion. All correlations are summarized in Table 4.

Correlation with Non-Emotional Facial Expression Ability. Non-emotional expression ability was indicated by five indicators (corrugator, levator, depressor, frontalis, and zygomaticus, as described earlier) and had a good fit (χ^2^(5) = 2, *p* = .789; CFI = 1; TLI = 1; RMSEA < .001; SRMR = .019). The factor was reliable with ω = .589. 

Next, we modeled M6b and non-emotional expression ability jointly and let the factors correlate. We expected and confirmed a large correlation between the general factors (r = .722, *p* < .001). 

Correlations with Receptive Ability Constructs. The receptive abilities were modeled in line with previously established measurement models ([44]; including the residual correlations added there). 

Facial emotion perception and recognition was indicated by seven variables, one for each task, had good fit (FEC model: χ^2^(14) = 16, *p* = .297; CFI = .994; TLI = .991; RMSEA = .026; SRMR = .028), and was reliable (ω = .781). When modeled with emotion expression ability, we found a medium correlation between the two abilities (r = .305, *p* < .001), supporting our hypothesis. 

For facial identity perception and recognition, we estimated one to two indicators per task (e.g., “sequential matching” was split into a “part” and a “whole” parcel), resulting in nine indicators total. The model had good fit to the data (χ^2^(26) = 76, *p* < .001; CFI = .942; TLI = .919; RMSEA = .090; SRMR = .063) and was reliable (ω = .766). Contrary to our expectations, facial identity perception and recognition had only a very small correlation with emotion expression ability (r = .150, *p* = .032).

Posture emotion recognition was modeled with parceled indicators built by combining DANVA-2 posture items with emotion-specific parcels (i.e., items with the same target emotion were pooled into a parcel). To achieve acceptable model fit (χ^2^(1) = 2, *p* = .163; CFI = .977; TLI = .860; RMSEA = .063; SRMR = .022), we added one residual correlation based on modification indices. The posture emotion recognition factor had a small factor saturation: ω = .401. As expected, we found a small correlation between emotion expression ability and posture emotion recognition (r = .273, *p* = .010). 

To model emotion management, we built four parcels from the STEM items. The model had a good fit, and the factor was reliable (χ^2^(2) = 3, *p* = .207; CFI = .993; TLI = .978; RMSEA = .049; SRMR = .020; ω = .705). In line with our hypotheses, emotion management correlated weakly with emotion expression ability (r = .196, *p* = .015).

Emotion understanding was indicated by five parcels, each composed of responses to the STEU items. The model had a mostly good fit and reliability (χ^2^(5) = 11, *p* = .047; CFI = .952; TLI = .904; RMSEA = .073; SRMR = .040; ω = .634). Emotion expression ability correlated weakly with emotion understanding (r = .184, *p* = .024). 

*G* was indicated by nine variables, namely, performance on fluid intelligence, working memory capacity, and immediate and delayed memory tests. The general mental ability model had a mostly acceptable fit (χ^2^(24) = 95, *p* < .001; CFI = .929; TLI = .893; RMSEA = .112; SRMR = .062), and the factor was very reliable (ω = .801). General mental ability had a small correlation with emotion expression ability (r = .224, *p* = .005).

Correlations with Extraversion. Extraversion was indicated by six variables, one for each facet of extraversion ([13]). The measurement model had bad model fit (χ^2^(9) = 76, *p* < .001; CFI = .847; TLI = .745; RMSEA = .177; SRMR = .074), but reliability was high with ω = .791. Insufficient fit is a common problem in modeling self-report questionnaires (see e.g., [74]). However, as extraversion is only a covariate in this study, we did not strive to optimize their models and instead followed through with a broad factor that is comparable with but superior to computing and using a manifest mean score. The extraversion factor had a weak correlation with emotion expression ability: r = .165, *p* = .025.

Joint Evaluation of Hypotheses. In the last step, we evaluated how closely the empirical correlations matched our predicted rank order of correlations, based on the expected effect sizes reported in Table 1. To evaluate this match, we estimated Spearman’s ρ. The correlation was ρ = .627, indicating a strong match of the predicted and empirical correlations. 

### 2.3. Conclusions

In our first study, we introduced new tests of emotion expression ability, tested competing measurement models of expression ability, and positioned this ability in a nomological network of related constructs. We found that three sources of variation are required to describe individual differences in performance on our emotion expression tasks. These are (1) a general factor of emotion expression ability, (2) task-specific variation, and (3) emotion-specific variation. 

The general factor of emotion expression ability correlated mostly as expected. We found medium to strong correlations with closely related socio-emotional abilities, weak to small correlations with more distantly related socio-emotional abilities, a small correlation with non-emotional cognitive abilities, and a weak correlation with extraversion. 

The correlational pattern hints toward an overarching socio-emotional abilities construct incorporating receptive and productive abilities. Furthermore, the correlations with general cognitive abilities extend the idea of a positive manifold. Overall, this is evidence that a broad socio-emotional abilities factor, including emotion expression ability, might be considered an additional second stratum factor in models of intelligence. We additionally learned that extraversion plays a role in successful emotion expression. 

In study 2, we strive to replicate these results with optimized emotion expression tasks (i.e., only imitation with feedback, with a complete trial list) and a reduced set of covariate tasks. In study 3, we will again replicate these findings and extend them to other abilities, including faking ability and crystallized intelligence. 

## 3. Study 2

### 3.1. Methods

#### 3.1.1. Sample

For study 2, we recruited n = 159 healthy participants from a university student participant database in Ulm, Germany. After excluding participants with incomplete data in the emotion expression task, the sample size was n = 141. Approximately half of the participants (51%) were female. Due to matching issues and incomplete responses, age, and education data were only available for n = 109 participants. For these, the mean age was *AM* = 23.42 (*SD* = 4.23), and 26% already had an academic degree. 

The largest SEM in this study had 25 indicators, so the sample size is sufficient to fulfill the five observations per indicator rule of thumb ([2]). With the same significance threshold (α = .05) and power (1 − β = .80) as in study 1, we have enough sensitivity to detect bivariate manifest correlations as small as r = |.21| in study 2 (calculated with G*Power; [31]). No data from this study used in this manuscript have been published elsewhere. 

#### 3.1.2. Procedure and Measures

Study 2 consisted of two parts: (1) a lab study and (2) online questionnaires and a demographic survey. Data were matched anonymously with individual participant codes, but from the n = 141 participants, 22 participants did not report the same code in both parts and could not be matched. Consequently, part 2 data are only available for n = 119 participants. We will only discuss the tasks relevant to this paper. In terms of the order of tasks, emotional and non-emotional tasks were alternated, if possible. For practical reasons, i.e., the camera recording, the expression tasks were presented consecutively. The facial emotional stimuli used across different tasks never had an identity overlap across tasks. This means no two tasks with such stimuli contain stimuli from the same person. Please note that several tasks in part 1 were about pain sensation and regulation, which are not reported in this paper. All tasks in part 1 were programmed in and presented through Inquisit 4 ([23]).

Expression Tasks. The lab study included two emotional expression ability tasks (production and imitation with feedback) and the non-emotional expression ability task, as they are described in Step I. 

Facial Emotion Perception and Recognition (FEPR). Facial emotion perception was measured with the “identification of emotion expressions from composite faces” task from the BeEmo test battery by [119] ([119]). In this task, participants must identify facial emotion expressions with six basic emotion labels in either the top or bottom halves of composite faces expressing different emotions in the top and bottom halves. The task consists of 72 items and is highly reliable (ω = .81). According to the recommendations of the original authors, we scored the task with unbiased hit rates ([115]). Facial emotion recognition was measured with the “cued emotional expressions span” from the emotion perception and recognition task battery by [119] ([119]). In this task, participants learn three to six expressions of one person with varying intensities in one block and then recall them by identifying them among distractors. The task consists of seven such blocks and is reliable (ω = .59, [119]). We calculated the percentage of correctly remembered faces per block.

General Mental Ability. As a proxy to *g*, fluid intelligence was measured with the figural task of the Berlin Test of Figural and Crystallized Intelligence (Berliner Test zur Erfassung von Fluider und Kristalliner Intelligenz, BEFKI-gff; [121]), version 8-10. In this task, participants must complete sixteen sequences of figural images via logical deduction of rules explaining the change in the displayed sequence. Items are ordered by difficulty from lowest to highest. The task is well-established and very reliable with ω = .87.

Extraversion. Extraversion was assessed with the NEO-FFI ([3]), a well-established and reliable (α = .81) questionnaire to measure the Big Five personality factors on a factor level. 

### 3.2. Results

#### 3.2.1. Step II: Measurement Models of Facial Emotion Expression Ability

As in study 1, parcels for the expression tasks were calculated based on the respective muscle origin for the non-emotional expression task (e.g., one *m.zygomaticus* parcel) or based on emotion sets (e.g., one anger in production parcel). This resulted in the same 12 emotional expression indicators, which were used to test the measurement models M1 to M6, as displayed in Figure 2, and five non-emotional expression parcels.

We tested models M1 to M6 (see Figure 2) to determine the best-fitting measurement model of emotion expression ability (summarized in Table 3). Both Models M5 and M6 had a good fit, with M6 fitting the best. Therefore, we again concluded that M6 had the best fit to the data and that all three sources of variation, namely, a general factor, task-specific variation, and emotion-specific variation, need to be distinguished to account for the data. Because the model again had convergence problems when relating model M6 with the covariates, we used model M6b, which replaces the emotion-specific factors with residual correlations of the same emotion indicators. M6b had good model fit (χ^2^(42) = 46, *p* = .327; CFI = .992; TLI = .987; RMSEA = .024; SRMR = .053). The general emotion expression factor was reliable (ω = .639), and the imitation factor had low reliability (ω = .374).

#### 3.2.2. Step III: Nomological Network 

Before correlating emotion expression ability with its covariates, we first developed measurement models of the covariates to guarantee factorial validity and ensure sufficient factor saturation of covariates. As in study 1, each covariate in the nomological network was modeled and correlated with emotion expression ability independently. We additionally modeled the receptive abilities using a bifactor approach; those results are available in the Appendix A. 

As in study 1, we established emotion expression in a nomological network of socio-emotional traits through three steps: 1. correlation with non-emotional expression; 2. correlation with receptive abilities; and 3. correlations with extraversion (see Table 5).

Correlation with Non-Emotional Facial Expression Ability. Non-emotional expression ability was modeled as in study 1 with the same strategy for parceling. After allowing one residual correlation between indicators of mouth-related muscles, the zygomaticus, and the depressor indicator, the model had a (mostly) good fit to the data (χ^2^(4) = 7, *p* = .158; CFI = .956; TLI = .891; RMSEA = .068; SRMR = .038), and the factor was reliable (ω = .609). When correlating general emotion expression with non-emotional expression ability, we found a large correlation (r = .816, *p* < .001), thus supporting our hypothesis.

Correlations with Receptive Abilities. Facial emotion perception and recognition was modeled based on the six unbiased hit rate scores (one per basic emotion) from the facial emotion perception task and the seven average scores per block from the facial emotion recognition task. To achieve a good model fit, modification indices indicated that in addition to a general factor loading on all indicators, an orthogonal factor loading on the seven facial emotion memory indicators from the “cued emotional span” task was required. This model had a good fit (χ^2^(58) = 36, *p* = .989; CFI = 1; TLI = 1; RMSEA < .001; SRMR = .045), but the general facial emotion perception and recognition factor’s reliability was low (ω_FEC_ = .16). This factor had a medium correlation with emotion expression ability (r = .347, *p* = .017).

*g* was modeled based on four sequential parcels by averaging items 1, 5, 9, and 13; items 2, 6, 10, and 14; etc., from the BEFKI-gff. The model had a good fit (χ^2^(2) = 1, *p* = .492; CFI = 1; TLI = 1; RMSEA < .001; SRMR = .012) and high reliability ω = .828. Fluid intelligence correlated weakly with emotion expression ability (r = .107, *p* = .177).

Correlations with Extraversion. Extraversion was modeled as a general factor loading on all 13 NEO-FFI extraversion items. The model had a (mostly) bad fit (χ^2^(65) = 181, *p* < .001; CFI = .766; TLI = .719; RMSEA = .122; SRMR = .096) but high reliability (ω = .830). As discussed in study 1, because self-report measures are often prone to bad model fit, we did not try to optimize the model. When correlated with emotion expression ability, we found a small correlation (r = .205, *p* = .046).

Joint Evaluation of Hypotheses. We calculated Spearman’s ρ to evaluate how closely the rank order of empirical correlations resembled the expected rank order of correlations in the nomological network, as described in Table 1. We found a correlation of ρ = .800, indicating a very good match of rank order. 

### 3.3. Conclusions

For our second study, we updated the emotion expression ability test compared to study 1 with complete trial lists and focused our assessment of imitation to just the imitation with feedback task. Based on this adjusted item set, we again calculated 12 emotion- and task-specific parcels (as in study 1) and tested the same measurement models. We found that the same model (M6) representing three sources of variation (general ability, task-specific, and emotion-specific) fit the data best. 

Next, we estimated correlations between emotion expression ability and non-emotional expression ability, facial emotion perception and recognition, non-emotional cognitive abilities, and extraversion. We replicated the very strong correlation with non-emotional expression ability found in study 1. Furthermore, we found emotion expression ability was moderately related to receptive emotional abilities, which supports the idea of positive manifold prediction among socio-emotional ability constructs. This effect must be interpreted with caution because the facial emotion perception ability factor had very low reliability. Emotion expression ability was also weakly related to indicators of general intelligence, supporting the idea that the construct our tasks assess can be considered a socio-emotional ability and that socio-emotional abilities can be considered intelligence. Additionally, we again found a small relation to extraversion, stressing the latter’s importance in successful emotion expression.

We conclude that the findings from study 1, namely, that emotion expression ability is a socio-emotional ability that can be measured via maximal performance tests and that fits well in a nomological network of socio-emotional traits, could conceptually be replicated in this study. In our last study, we will again strive to replicate the measurement model of emotion expression ability in a more applied sample and extend the nomological network with new abilities, namely, faking ability and crystallized intelligence.

## 4. Study 3

### 4.1. Methods

#### 4.1.1. Sample

The n = 123 participants from Ulm, Germany, recruited for this study were all about to finish an academic degree or had recently finished an academic degree. Demographic data were missing for two participants. The rest of the sample was 57% females and had a mean age of *AM* = 25.15 (*SD* = 3.54); 36% were at their Bachelor’s level, 51% at their Master’s, 5% at their PhD level, and 8% did not report their current educational level.

As the largest SEM in this study consisted of 21 indicators, the sample size is sufficient to fulfill the five observations per indicator rule of thumb ([2]). With the same significance threshold (α = .05) and power (1 − β = .80) as in both prior studies, we have enough sensitivity to detect bivariate manifest correlations as small as r = |.22| in study 3 (calculated with G*Power; [31]). No data from this study used in this manuscript have been published elsewhere. 

#### 4.1.2. Procedure and Measures

While in studies 1 and 2, we collected data in regular lab settings, in our third study, we wanted to measure emotion expression ability in a more applied setting. Thus, the data were collected in a study about assessment interviews. All measures were presented within a simulated job assessment situation for students about to apply for their first job outside of university. This study involved three parts: part (1) online questionnaires, part (2) cognitive testing portion, including cognitive testing, and part (3) a simulated job interview, for which participants received personalized feedback as a reimbursement for their participation in this study. Data from the interview are not reported in this manuscript but are reported elsewhere ([72]). 

Participants completed the emotional expression battery (production and imitation with feedback) and non-emotional expression task, as described earlier, in addition to measures of facial emotion perception and recognition, faking ability, general mental ability, and crystallized intelligence. Faking ability tasks were administered in part 1, whereas all other tasks were administered in part 2 using Inquisit 4 ([23]). In terms of the order of tasks, emotional and non-emotional tasks were alternated, if possible. For practical reasons, i.e., the camera recording, the expression tasks were presented consecutively. Facial emotional stimuli used across different tasks never had an identity overlap across tasks. This means no two tasks with such stimuli contain stimuli from the same person.

Facial Emotion Perception and Recognition (FEPR). Participants completed one test of facial emotion perception (“identification of emotion expressions of different intensities from upright and inverted dynamic face stimuli”) and one test of facial emotion recognition (“memory game for facial expressions of emotions”) from the BeEmo battery ([119]). In the “upright-inverted” task, participants see short dynamic facial emotional expressions either presented upright or inverted and must label them with one of the six basic emotions. We scored the task as recommended with six unbiased hit rates ([115]), one per emotion. The task is reliable with ω = .62. In the “memory game”, participants play the well-known game “memory”, where they must find picture pairs, with pictures of facial emotion expressions. The game consists of four blocks with three, six, and twice with nine pairs. We calculated the mean correct responses per block but excluded the first block due to extreme ceiling effects. The task is reliable with ω = .60. We chose different tasks than in study 2 to allow us to further generalize the findings in the case of a conceptual replication.

Faking (Good) Ability (FGA). Faking ability refers to the socio-emotional ability to adjust responses in self-report questionnaires to present a response profile that matches a certain goal, such as the goal to get hired in a job assessment ([35]). Participants were asked to fake six job profiles using the Work Style Questionnaire (WSQ, [4]), namely, commercial airplane pilot, TV/radio announcer, tour guide, software developer, security guard, and insurance policy processing clerk (the first three were also used in [35]). The responses for each job were each scored with the profile similarity index shape, essentially a correlation between a participant’s response profile and the optimum response profile for a given job (see [35], for details on scoring). While a version of this task design had low reliability (ω = .33), it had strong validity indicated by significant correlations with general cognitive abilities, crystallized intelligence, and facial emotion perception and recognition reported elsewhere ([35]).

General Mental Ability (*g*). In this study, *g* was indicated by fluid intelligence assessed with a verbal deduction test, the verbal task of the BEFKI (BEFKI-gfv; [97]), version 11-12+. Participants solved 16 items presented in short verbal vignettes with relational reasoning. Items are ordered by difficulty from lowest to highest. The task is well-established and very reliable (ω = .76). 

Crystallized Intelligence (*gc*). Crystallized intelligence was assessed with a shortened version of the BEFKI-gc ([97]). Participants completed short general knowledge multiple-choice questions drawn from the domains of natural sciences, social sciences, and humanities. The task was shortened for this study from 64 to 32 items by excluding items with redundant item difficulties. The original version is reliable (Dimitrov’s ρ = .83, Schipolowski et al. 2020).

### 4.2. Results

#### 4.2.1. Step II: Measurement Models of Facial Emotion Expression Ability

The parcels for the expression tasks were calculated as in study 2, which is 12 emotion expressions. We tested the same six pre-specified measurement models of emotional expression (see Figure 2) as in studies 1 and 2. The results are summarized in the right part of Table 3. As in studies 1 and 2, M6 had the best fit. For consistency, we instead used model M6b (χ^2^(42) = 55, *p* = .093; CFI = .968; TLI = .949; *RMSEA* = .049; *SRMR* = .055) to estimate correlations with other constructs. The general factor of emotion expression ability had good factor saturation (ω = .718), while the imitation factor saturation was substantially lower (ω = .428).

#### 4.2.2. Step III: Nomological Network

As in the two prior studies, to ensure factorial validity, we developed individual measurement models for each covariate construct before correlating the constructs with emotion expression ability. Each measurement model was based on parcels of items or composite task scores (see the Appendix A for illustrations of these models). We additionally modeled the receptive abilities using a bifactor model; those results are also reported in the Appendix A. All correlations are summarized in Table 6.

Correlation with Non-Emotional Facial Expression Ability. Non-emotional expression ability was again indicated by five parcels, calculated the same as in studies 1 and 2. To achieve a good model fit for the non-emotional expression task, we had to allow one residual correlation between the Levator and Depressor parcels. This model had a (mostly) good fit (χ^2^(4) = 4, *p* = .360; CFI = .979; TLI = .948; RMSEA = .027; SRMR = .038) and rather low reliability (ω = .415). As in studies 1 and 2, non-emotional expression had a strong correlation with emotional expression (r = .762, *p* < .001), supporting our hypothesis.

Correlations with Receptive Ability Constructs. Facial emotion perception and recognition was modeled based on six unbiased hit rate scores from the facial emotion perception task and three block-based indicators (after excluding the score of block 1 due to ceiling effects) from the facial emotion recognition task. Facial emotion perception and recognition was modeled similarly to the respective model in study 2: a general factor loading on all parcels from both receptive emotional tasks and a nested orthogonal factor loading on the three “memory game” parcels. This facial emotion perception and recognition model had a good fit (χ^2^(25) = 28, *p* = .295; CFI = .987; TLI = .981; RMSEA = .033; SRMR = .041), and the general factor was very reliable (ω = .809). Facial emotion perception and recognition had a smaller correlation with emotion expression ability than in the previous studies (r = .110, *p* = .173).

As indicators for the faking good ability factor, we used the shape scores per faked job, delivering six indicators. To achieve a good model fit, we had to allow one residual correlation between the security guard and the insurance clerk parcel (χ^2^(8) = 6, *p* = .669; CFI = 1; TLI = 1; RMSEA < .001; SRMR = .035). The faking factor was reliable (ω = .581) and, in a separate model, weakly related to emotion expression ability (r = .170, *p* = .109).

Crystallized intelligence was indicated by three BEFKI-gc scores calculated based on the three science domains, namely, all items belonging to natural sciences were combined into a natural science indicator, all items from the social science scale were combined into a social science indicator, and all items from the humanities domain were combined into a humanities indicator by averaging across the respective set of items. The model was just identified, indicating a perfect model fit. The factor had a good reliability (ω = .763). The correlation between emotion expression ability and crystallized intelligence was weak (r = .164, *p* = .060).

*g* was modeled based on parceled BEFKI-gfv scores with the same parceling logic as the BEFKI-gff in study 2: sequential selection of four items each to summarize them in a parcel. The model fit was good (χ^2^(2) = 2, *p* = .458; CFI = 1; TLI = 1; RMSEA < .001; SRMR = .020), and the reliability was high (ω = .704). The correlation between fluid intelligence and emotion expression ability was effectively zero (r = .088, *p* = .238).

Joint Evaluation of Hypotheses. Using Spearman’s ρ, we evaluated how closely the rank order of empirical correlations in this study matched the rank order of expected correlations based on Table 1. With the overall lower correlations in this study, the rank order fits a little worse than in studies 1 and 2 but still well ρ = .616.

### 4.3. Conclusions

Our third study replicated the measurement models of emotion expression ability with data from a more applied setting. We again found that the complex measurement model M6, which considers three sources of variation (general ability and task- and emotion-specific) fit the data best. We additionally replicated relations to other socio-emotional and non-emotional abilities, and extended the nomological network of the construct, showing additional relations to another complex socio-emotional ability, faking good ability, and another relevant non-emotional ability, crystallized intelligence. 

The correlations with non-emotional expression and non-emotional receptive cognitive abilities were (except for fluid intelligence) as expected (and as in earlier studies) and the correlation with faking good ability was weak, but only slightly lower than expected. The correlation between facial emotion perception and recognition, however, was smaller than expected. Given that in the other studies, and specifically in study 1, which had multiple tasks for the construct, we found the expected effects, this low correlation might be explained by the task-specific method’s effect on the facial emotion perception and recognition tasks used here. 

Overall, the correlations matched the expected rank order of correlations in the nomological network fairly well. Thus, we can conclude that emotion expression ability can be considered a valid construct that fits very well in a nomological network of socio-emotional abilities and can be incorporated into models of intelligence.

## 5. General Discussion

### 5.1. Summary and Interpretation of Results

With this paper, we strive to establish the ability to pose emotional expressions in the face, a relevant ability that has been ignored relative to receptive emotional abilities ([27]), in a nomological network of socio-emotional traits and intelligence. We followed three steps: I. Developing an aptitude test of emotion expression ability and an objective scoring procedure for such a test. II. Establishing a measurement model of emotion expression ability and isolating sources of variation in performance. III. Evaluating correlations with other constructs in a nomological network of socio-emotional traits and intelligence and testing their rank order against a theoretically deducted rank order of correlations in this nomological network.

#### 5.1.1. Step I: Task Development

Going beyond prior measurement approaches to emotion expression abilities that did not fulfill all criteria of aptitude testing of this construct, we introduced new tests to measure the ability to pose facial emotion expressions. The tests capture the relevant behavior as a response, namely, facial expressions—instead of responses to item statements in self-report questionnaires—and instruct participants to pose emotions to their best ability. The expressions are scored according to their communicative value by objective computer software that outperforms human raters ([10], [9]; [33]). Considering the communicative function of emotion expressions, and that the more prototypically and intensely an expression is, the more it maximizes the chances of being perceived correctly, we consider our test scores to be veridical. In sum, we demonstrated that emotion expression ability can be measured objectively and adhering to standards of aptitude testing ([15]).

#### 5.1.2. Step II: Measurement Model

Across three studies, we demonstrated that to explain individual differences in emotion expression, three major sources of variation must be considered in a measurement model: (1) the general ability to pose emotion, (2) task-specific variation (i.e., whether a task is a production or an imitation task), and (3) emotion-specific variation (i.e., which emotion is to be expressed). In all three studies, the measurement model M6 that incorporated all three sources of variation was superior in fit and reached acceptable to mostly good fit levels. 

This is crucial because it has implications on how tests of emotion expression ability should be treated in research and application. The complexity of the measurement model demonstrates that a univariate assessment of this ability is not sufficient and instead, multiple tests possibly involving multiple target emotions should be used to measure it properly. If possible, with regard to sample size limitations, latent factor models representing the complexity of the construct are preferable over manifest scores that cannot dissect the different sources of between-subject variation in emotion expression ability. 

Although complex, the final measurement model also includes a general factor that represents a general ability to pose emotional expressions. But the nature of the imitation-specific and emotion-specific variance is less than clear. For example, in models of psychopathy, imitation is argued to be a specific ability in which psychopaths should perform better compared with production (for a summary of this hypothesis, see [78]). However, the factor could also represent method variation due to the different stimuli and does not contain any relevant trait variance. Given this factor had consistently lower reliability than the general factor and our limited sample sizes, we could not further explore the nature of this factor. Future studies with larger samples, more tasks, and more task types will hopefully add to our understanding of task-specific variance in emotion expression ability.

Similarly, the emotion-specific variation may be attributable to something meaningful, such as individual differences in adherence to display rules of emotions (e.g., [68]), that is, individual differences in typical emotion expression behavior. But it could also represent scoring artifacts in expression software, e.g., range differences between emotion scores ([10], [9]) might be indicative of differences in discriminatory power of scores.

Since exploring the nature of these specific factors was not the goal of this research, our study designs did not allow us to test these hypotheses. Yet, we hope that future studies address these research questions. For example, including more imitation tasks (e.g., dynamic imitation, imitation of emojis, etc.) would allow us to compare multiple task-specific factors against one overarching imitation factor. Whatever solution fits the data best could then be correlated with other traits or criteria to test the factor’s (or factors’) validity. Similarly, more tasks (preferably in larger samples than ours) would allow us to model stable emotion-specific factors and test their validity. Alternatively, existing video data could be reanalyzed with other expression recognition software to see whether the emotion-specific variation replicates.

#### 5.1.3. Step III: Nomological Network

Given that we found a stable general factor of emotion expression ability in all three studies, we could embed this factor in a nomological network of socio-emotional traits and intelligence. We tested correlations with non-emotional posing, receptive socio-emotional abilities, general cognitive abilities, and extraversion, and we found substantial support that emotion expression is a viable extension to a socio-emotional abilities factor in models of intelligence and correlates weakly with extraversion.

Specifically, emotion expression ability was strongly correlated with non-emotional expression ability, suggesting that the ability to express emotions is built upon a more basic ability to control one’s facial muscles, presumably due to the shared neural circuit. While the two abilities are strongly related, they are still distinct and share only 52% to 67% of their variance. This finding is comparable to the relation between emotional and non-emotional face perception and recognition tests ([44]). We conclude that there is a distinct ability to pose prototypical emotional expressions that is different from basic facial muscle control. Yet, facial muscle control might be a limiting factor to emotion expression abilities. Some faces might be more prone to expressing emotions than others, i.e., they have more facial plasticity. Thus, studies on emotion expression abilities should always also include measures of basic facial muscle control. Future research could further explore the specificity of emotion expression ability, e.g., by including culturally known expressions that are not directly related to a basic emotion (e.g., winking) in new tests.

With regard to receptive socio-emotional abilities, we expected emotion expression ability to correlate medium to large with facial emotion perception and recognition, the perceiving part in the emotion communication channel. In two studies, correlations with emotion perception and recognition were of medium to small size, i.e., slightly smaller than expected for facial emotion perception and recognition. This and other correlations were somewhat smaller in study 3, which might be explained by weaker representations of covariates. Overall, though, our findings replicated earlier findings ([28]) and extended them, especially with the multivariate study design of this study. The findings are meaningful evidence for an overarching factor of socio-emotional abilities. We recommend that test batteries of emotional intelligence, which strive to offer a broad assessment of socio-emotional abilities, include tests of emotion expression ability in the future. 

Because facial emotion and identity perception and recognition are closely related ([44]), we expected only slightly lower correlations between emotion expression and facial identity perception and recognition than we found for facial emotion perception and recognition. The correlation we found was lower than expected, indicating that it is shared emotional content that is crucial to explain the communality between emotion expression ability and other emotional abilities. In other words, emotional abilities seem to share some specific variation beyond other face-related abilities that are worth further investigation. This idea was further supported by the very small correlations we found with other distant emotional abilities: emotion management and emotion understanding. We would expect these correlations to increase if the latter abilities could be measured with actual ability tests and not just SJTs. Nevertheless, correlations with non-emotional social abilities (i.e., identity perception and recognition, faking good) were still sufficient in our studies, hinting toward a general socio-emotional abilities factor beyond mere emotional intelligence.

In sum, the correlational network across socio-emotional abilities found in our studies can be considered evidence that all these abilities form an overarching socio-emotional abilities factor that might be considered a stratum II construct in the Cattell–Horn–Carroll (CHC) model of intelligence. To support this idea, a tentative stratum II construct must demonstrate a positive manifold, the phenomenon that all intelligence tests correlate ([104]). Therefore, we also tested the correlation between emotion expression ability and other general cognitive abilities, such as fluid intelligence, working memory capacity, immediate and delayed memory, and crystallized intelligence, and we found small and meaningful correlations. Combining our results with earlier work demonstrating correlations between our covariates and general cognitive abilities ([35], [34]; [43], [44]; [66]; [65]; [76]; [100], [101]) supports the hypothesis that socio-emotional abilities can extend the CHC model with a new stratum II construct.

Lastly, correlations with personality traits can inform us further about the nature of new constructs. For example, some typical behavior assessed in personality questionnaires might help to improve an ability throughout a lifetime, such as more open people tend to have slightly higher cognitive abilities ([20]; [36]). When it comes to socio-emotional abilities, it can be argued that people with high extraversion engage in more social interaction, which gives them more opportunities to develop socio-emotional abilities. In fact, we found a consistent, small correlation between extraversion and emotion expression ability in studies 1 and 2, supporting this hypothesis. We conclude that extraversion can be a crucial factor in understanding individual differences in socio-emotional abilities, which might inform possible interventions and training. Furthermore, as one of the reviewers pointed out, it would be interesting to further explore the link of maximal performance emotion expression ability with typical emotion expression behavior. We hope that future studies can, for example, explore the hypothesis that individuals with typically more expressive behavior also perform better in emotion expression ability tasks.

### 5.2. Implications

We demonstrated how facial emotion expression-posing ability can be measured objectively and, according to standards of aptitude testing, how it should be modeled, and that this ability is a valid socio-emotional ability by testing correlations in a nomological network of socio-emotional traits and intelligence. Indeed, our findings help to manifest socio-emotional abilities as a second-stratum intelligence factor. We believe this can open many gates to new research in many different psychological fields. 

The task battery of emotion expression tasks might be further extended with other target modalities than words or faces, such as emojis or emotionally laden color slides, as in the slide-viewing technique ([8]). Furthermore, new tasks could capitalize on the dynamic nature of emotion expression and present dynamic targets or have participants follow a dynamic series of expressions across different emotions. Interestingly, new studies in this field are becoming easier as facial emotion expression recognition becomes more and more common in hand-held devices, which will allow for emotion expression to be tested at home or on the fly ([33]). This would allow us to test more complex measurement models of emotion expression abilities. 

The fact that testing emotion expression abilities becomes easier with handheld devices is also a great chance for more applied or contextualized research such as ambulatory studies ([110]). Noticeable problems in emotion expression are a part of many psychological disorders, such as autism ([77]). However, to our knowledge, the relation between objective aptitude tests of emotion expression and the nature and intensity of mental disorders has not been studied yet. If emotion expression is a key feature of several mental disorders, the procedures outlined here might help to capture deficiencies in this ability. Beyond asking participants to express emotions as strongly as they can—obviously, it would be exciting to facilitate ambulatory assessment to learn about the role of typical emotion expressions in a real-life context—typical expressions might be used to inform about affective states and help to establish early warning systems for mood disorders. Daily selfies might be used to understand individual differences in expressive ranges and how these interact with people’s social environment. Similarly, regular emotion expression ability measurements via ambulatory assessment might add valuable information to understanding people’s states.

Furthermore, performance appraisals in a recruitment context might benefit from emotion expression tasks. Clearly, emotion expression ability might be beneficial in many jobs that require high extraversion, such as retail and sales, or in jobs in the service sector that include face-to-face interaction. Given that emotion expression ability is only weakly related to general mental ability (*g*), the most prominent predictor of job performance and success, emotion expression ability is a promising candidate to deliver incremental predictive validity in job assessments.

For the same reason, it might be a promising candidate to predict other relevant real-life outcomes. Emotional intelligence is reported to predict academic performance ([67]) and emotion regulation ability predicts income and well-being ([14]), so the hypothesis that emotion expression ability could, too, is not far-fetched. Lastly, if emotion expression ability was demonstrated to be a relevant predictor of real-life outcomes, it could also be a promising candidate to be improved by training. While many convincing training studies show only very small overarching effects (general mental abilities: [102]; face identity perception and recognition: [22]), because the ability to be trained is just an all too common ability used throughout life so often that everyday life has already maxed out individual capacity, emotion expression ability is presumably less often used and might still offer potential for improvement. Existing smartphone apps, such as Emotion Hero ([113]) could easily be used as a blueprint to develop training apps for emotion expression ability and test this hypothesis. 

### 5.3. Limitations

Although we see great potential in the task presented here and its validity, some limitations restrict the interpretation of our results. First, the correlations with receptive tasks were somewhat smaller than expected and sometimes not significant. Missing significance might be attributed to the smaller sample sizes of studies 2 and 3 and the reduced statistical power to reliably identify smaller effects. Given that across all studies, relations between emotion expression ability and other constructs were mostly replicated, we consider our findings trustworthy. Some of the hypothesized relations were weaker than expected. This attenuation might be explained by the productive versus receptive nature of expression tasks on one side and perception and recognition tasks on the other side. 

Relatedly, the reliabilities of some factors are only moderate or even low, highlighting that these factors should not be used for single-person assessments, as for these, good reliabilities (e.g., Cronbach’s α or factor saturation ω > .70) are required. Furthermore, low reliabilities highlight the need for analyses that consider disattenuation procedures ([105]). To address this issue, we estimated all constructs as latent factors, which are already disattenuated ([51]), and used these factors to estimate correlations in our nomological networks. Future studies should attempt to improve the saturation of latent variables with low omegas.

This leads us to the second limitation, which is that the current study only covered one of four earlier-introduced expressive abilities, namely, posing. Enhancement, neutralization, and masking were not covered because they would require the experimenter to have control over the emotional state of participants. While this is complicated, ethically precarious, and presumably not even possible for basic emotions, it is achievable, for example, with pain, where pain sensitivity can be measured psychophysically. Consequently, future studies might expand our posing tasks to other expressive abilities by asking people to manipulate their facial expressions while experiencing pain. Such tasks would be expected to share some of the same method variation that the posing tasks have, especially enhancement, and could thereby help to further understand the construct(s) of expressive abilities.

Similarly, although we sought to cover a broad range of related constructs in our nomological networks throughout the studies, they clearly do not contain all possibly related constructs. For example, we had neither vocal emotional abilities ([99]), emotional knowledge ([98]), emotional creativity ([116]), deception detection abilities ([122]), nor the ability to judge other people’s emotions ([49]), just to name a few. However, with the emotion expression tasks presented in this study and the availability of the material, future studies will easily be able to fill these research gaps.

Lastly, our two emotion expression tasks required many factors to explain individual differences in them. At the same time, our sample sizes and test design did not allow us to further investigate task- and emotion-specific variation. Future studies should strive to test more and new posing tasks (see above) with larger sample sizes to allow for manifesting an understanding of this ability and other sources of variation in the respective tasks. For this upcoming research, the present psychometric arrangement for capturing emotion posing might be useful as a blueprint.

### 5.4. Conclusions

The ability to voluntarily express an emotion in the face while no emotion is felt, or emotion posing, is a highly relevant ability for communication in humans because we are social and rational animals. Nevertheless, this ability was previously largely understudied due to a lack of objective aptitude tests to assess individual differences in it. In this paper, we presented how this ability can be measured objectively by using state-of-the-art computer software to score facial expressions and adhering to standards of aptitude testing, such as veridical responses. Individual differences in the here-presented measurement approach could be explained by three sources of variation: a general posing ability and task-specific and emotion-specific variation. The general posing ability factor showed expected relations to other socio-emotional traits and intelligence across three studies. We conclude that the ability to pose emotion is a viable extension to a possible socio-emotional abilities factor in models of intelligence and see great potential for this construct to enhance our understanding of human communication, models of human intelligence, assessments in clinical and industrial psychology, and the prediction of real-life outcomes in future studies.

## Figures and Tables

**Figure 1 jintelligence-12-00027-f001:**
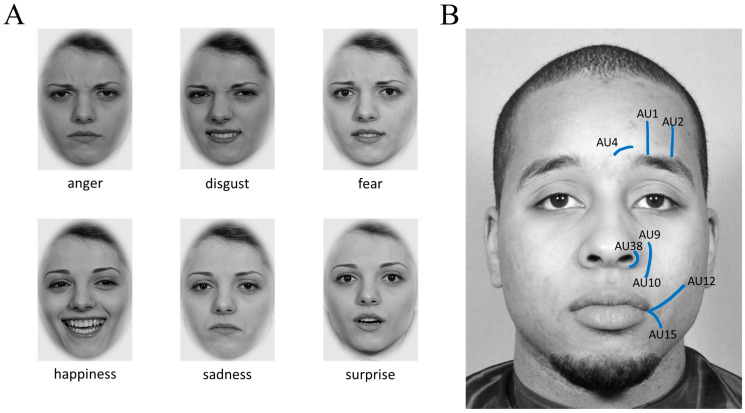
Target facial expressions that had to be shown during the posing tasks. (**A**): Six basic emotional expressions for the emotion-posing tasks. (**B**): Action units related to the movements in the non-emotional-posing task.

**Figure 2 jintelligence-12-00027-f002:**
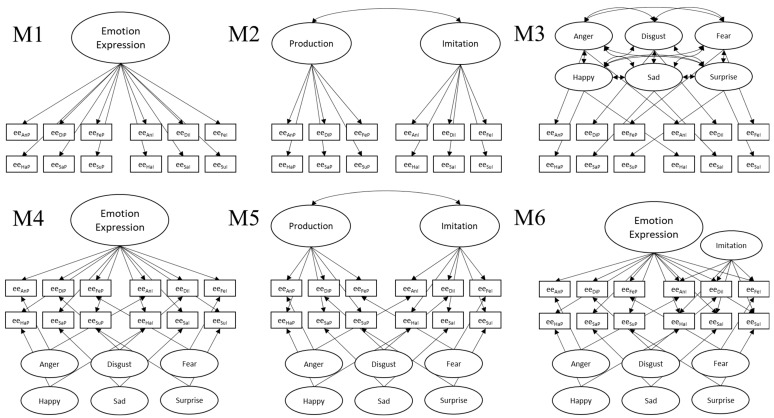
Measurement models of emotion expression-posing ability. EE = emotion expression; An = anger; Di = disgust; Fe = fear; Ha = happiness; Sa = sadness; Su = surprise; P = production; I = imitation.

**Table 1 jintelligence-12-00027-t001:** A summary of covariates across our studies, their methodological overlap with facial emotion expression ability, and their expected correlation with the latter.

Covariate	Definition	Theoretical Considerations on Relations to Emotion Expression Ability	Methodological Overlap					$\hat{r}$	Results: Observed Effect Size		
			A	SE	P	F	Emo		Study 1	Study 2	Study 3
Non-emotional expression ability	Ability to move facial landmarks independent of emotion	Shares the same neural system to produce facial expressions	X	X	X	X		Very large	Very large	Very large	Very large
Facial emotion perception and recognition (FEPR)	Ability to perceive, distinguish, learn, and recall facial identities	Receptive part of facial emotional communication	X	X		X	X	Medium to large	Medium	Medium	Very small
Facial identity perception and recognition (FIPR)	Ability to perceive, distinguish, learn, and recall facial emotion expressions	Shares broader neural network of facial information processing	X	X		X		Medium	Weak	-	-
Posture emotion recognition (PER)	Ability to perceive and distinguish posture emotion expression	Receptive part of emotion communication	X	X			X	Medium	Small	-	-
Faking good ability (FGA)	Ability to distort responses to personality questionnaires in order to portray a desirable personality	Just as posing a deceptive ability	X	X				Small to medium	-	-	Weak
Emotion management (EM)	Ability to regulate own’s and others’ emotions	Posing emotions is an expressive emotion management ability	X	X				Small	Weak	-	-
Emotion understanding (EU)	Ability to understand emotions in self and others	Posing requires emotion understanding	X	X				Small	Weak	-	-
Crystallized intelligence (*gc*)	Accumulated skills and knowledge	Posing requires (emotion) knowledge	X					Small	-	-	Weak
General mental ability (*g*) indicated by fluid intelligence (*gf*)/working memory capacity (WMC)/Immediate and delayed Memory (IDM)	gf: reasoning ability WMC: capacity of information units stored and handled in the working memory IDM: learning and recall of information	Spearman’s positive manifold: all cognitive abilities relate	X					Small	Small	Weak	Zero
Extraversion (E)	Outgoing, social, and active typical personality	High E gives more real-life practice for socio-emotional abilities		X				Weak	Weak	Small	-

*Note*. Methodological overlap acronyms are A = regular ability; SE = socio-emotional; P = productive ability; F = facial content; Emo = emotional content. $\hat{r}$ = theoretically expected correlational effect size. The covariates are ordered from highest to lowest conceptual overlap and expected correlation with facial emotion expression ability. Correlation effect size categories correspond to the following ranges: zero: <.100; weak: .100–.199; small: .200–.299; medium: .300–.499; large: .500–.699; very large: >.700. Effect sizes refer to disattenuated correlations. X indicate checkmarks.

**Table 2 jintelligence-12-00027-t002:** A summary of facial expression-posing ability tasks used throughout all studies.

Task	n_items_ in Study 1/2/3 (Excluding Baselines)	Example Item
Non-emotional production	24/12/12	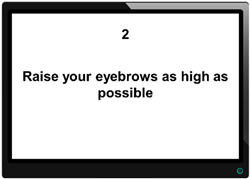
Emotional production	12/12/12	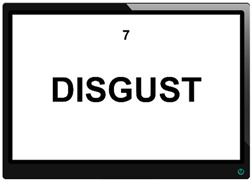
Emotional imitation without feedback	12/24/24	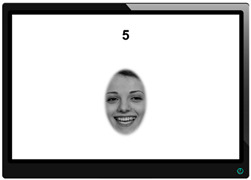
Emotional imitation with feedback	12/24/24	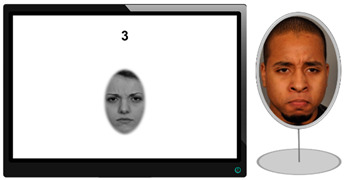

**Table 3 jintelligence-12-00027-t003:** A comparison of measurement models of facial emotion expression-posing ability across studies 1 to 3.

	Study 1; *n* = 237			Study 2; *n* = 141			Study 3; *n* = 123		
Model #	χ^2^(*df*); *p*	CFI/TLI	RMSEA/ SRMR	χ^2^(*df*); *p*	CFI/TLI	RMSEA/ SRMR	χ^2^(*df*); *p*	CFI/TLI	RMSEA/ SRMR
M1	χ^2^(54) = 448; *p* < .001	.572/.476	.175/.101	χ^2^(54) = 248; *p* < .001	.548/.448	.160/.103	χ^2^(54) = 180; *p* < .001	.678/.606	.137/.090
M2	χ^2^(53) = 447; *p* < .001	.571/.465	.177/.101	χ^2^(53) = 248; *p* < .001	.546/.435	.162/.103	χ^2^(53) = 179; *p* < . 001	.677/.597	.139/.090
M3	χ^2^(45) = 174; *p* < .001	.860/.794	.110/.067	χ^2^(45) = ; *p* < .001	.905/.861	.080/.084	χ^2^(45) = 78; *p* = .001	.914/.874	.078/.082
M4	χ^2^(47) = 170; *p* < .001	.866/.812	.105/.061	χ^2^(47) = ; *p* = .005	.932/.905	.066/.059	χ^2^(47) = 66; *p* = .023	.945/.923	.061/.056
M5	χ^2^(46) = 114; *p* < .001	.926/.894	.079/.058	χ^2^(46) = ; *p* = .176	.980/.971	.037/.059	χ^2^(46) = 61; *p* = .067	.961/.944	.052/.055
M6	χ^2^(41) = 91; *p* < .001	.946/.912	.072/.050	χ^2^(41) = ; *p* = .289	.989/.983	.028/.053	χ^2^(41) = 54; *p* = .076	.965/.944	.052/.055

*Note*. Models M1 to M6 are depicted in Figure 2.

**Table 4 jintelligence-12-00027-t004:** Study 1: Correlations between facial emotion expression-posing ability and other abilities and traits in the socio-emotional nomological network.

Covariate Category	Construct	r	*p*
Productive abilitiy	Non-emotional-posing ability	.722	<.001
Receptive abilities	Facial emotion perception and recognition (FEPR)	.305	<.001
	Facial identity perception and recognition (FIPR)	.150	.032
	Posture emotion recognition (PER)	.273	.010
	Emotion management (EM)	.196	.015
	Emotion understanding (EU)	.184	.024
	General mental ability (g)	.224	.005
Self-reported traits	Extraversion (E)	.165	.025

*Note*. Measurement models and fit for covariate constructs are described and reported in the Results Section of study 1 under III: Nomological Network. Models are displayed in the Appendix A. Correlations are estimated between latent factors (disattenuated). Unstandardized confidence intervals of covariances are reported in the results summary in the Appendix A.

**Table 5 jintelligence-12-00027-t005:** Study 2: Correlations between facial emotion expression-posing ability and other abilities and traits in the socio-emotional nomological network.

Covariate Category	Construct	r	*p*
Expressive ability	Non-emotional posing ability	.816	<.001
Receptive abilities	Facial emotion perception and recognition (FEPR)	.347	.017
	General mental ability (g/gf)	.107	.177
Self-reported trait	Extraversion (E)	.205	.046

*Note*. Measurement models and fit for covariate constructs are described and reported in the Results Section of study 2 under III: Nomological Network. Models are displayed in the Appendix A. Correlations are estimated between latent factors (disattenuated). Unstandardized confidence intervals of covariances are reported in the results summary in the Appendix A.

**Table 6 jintelligence-12-00027-t006:** Study 3: Correlations between facial emotion expression-posing ability and other abilities and traits in the socio-emotional nomological network.

Covariate Category	Construct	r	*p*
Expressive ability	Non-emotional posing ability	.762	<.001
Receptive abilities	Facial emotion perception and recognition (FEPR)	.110	.173
	Faking good ability (FGA)	.170	.109
	Crystallized intelligence (*gc*)	.164	.060
	General mental ability (*g*/*gf*)	.088	.238

*Note*. Measurement models and fit for covariate constructs are described and reported in the Results Section of study 3 under III: Nomological Network. Models are displayed in the Appendix A. Correlations are estimated between latent factors (disattenuated). Unstandardized confidence intervals of covariances are reported in the results summary in the Appendix A.

## Data Availability

The study materials, data, and analysis scripts for all three studies of this manuscript, as well as supplementary analyses and scoring syntax, can be accessed at https://osf.io/9kfnu/. Due to privacy restrictions of the facial stimuli, the imitation task is only available upon request from the corresponding author and after signing a user agreement. Because we do not hold the copyright of all covariate tests, all other measurement instruments used in the studies are not uploaded to a public repository, but they are available from the first author upon request or via the manuscripts in which they were originally published.

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
