# Peer review of "“Show Me What You Got”: The Nomological Network of the Ability to Pose Facial Emotion Expressions†"

_jintelligence, 2024, doi:10.3390/jintelligence12030027_

Round 1

Reviewer 1 Report

Comments and Suggestions for Authors

see attached document for specific comments

Author Response

Responses to the reviewer's points are reported in the attached file in blue.

Reviewer 2 Report

Comments and Suggestions for Authors

The current manuscript took on the difficult task of trying to assess emotion expression ability with the requisite measurement rigor to the standard required for aptitude testing.  After demonstrating that they had met that standard, the authors then went on to show this new measure was related as theoretically expected to other socio-metric abilities, including emotion receiving ability, emotion management, extraversion, and others.

This paper is well written, well executed, and definitely represents an original contribution to research in this area.  Moreover, there are clear extensions using this model that suggest compelling future research.

That said, there are a few areas that I feel need to be addressed before this manuscript is ready for publication. 

Literature Review: 

Unfortunately, a fair amount of the literature in this area often get siloed, but the claim that no one has attempted to create such a measure is not true.   The late Ross Buck did just that with his slide-viewing technique (SVT), and hit on many, if not all, of the features required for aptitude testing.  Further, he went on to show his measure of emotion expression ability was related to emotion receiving ability.  A good overview of this work is provided in his 1984 Communication of Emotion book.  Further, this slide-viewing technique was used in research on married couples and tested in Dave Kenny’s social relationship model paradigm – where it was shown that sending ability was considerably more important the receiving ability in successful communication in couples (See:  Sabatelli, Buck, and Kenny, 1986; Sabatelli, Buck, and Dreyer, 1982 and 1983).  There is also evidence that individuals who are veridical senders are better liked, probably overlapping with extraversion, but an independent effect of its own right (Sabatelli & Rubin, 1986).

In his work, Buck also highlighted the very different processes that underlie the spontaneous communication of emotion and symbolic communication of emotion which very much falls in the distinctions made in this paper.  Notably, the SVT focuses on the spontaneous communication of emotion, whereas the current effort focuses posed communication; a nice complement between the two efforts.

Methodology:

Given the number of tasks, particularly in various subareas, it is not clear what efforts were made to counterbalance order.  Theoretically, this are likely to be order effects, particularly if emotion sending and emotion receiving occur together.  The authors clearly tied production to perception – and thus these two activities are likely to influence each other.  What was done to mitigate the impact of this connection or other possible confounds?

Data analysis:  The data analysis is very sophisticated and lengthy.  Was there any effort to control for family-wise error given the number of tests?  Or, could the authors at least present an argument for why they were not concerned about this?

If I have tracked the full analysis correctly, it does appear that the most successful model (M6b) had broken the expressive scores out by emotion.   Further, on page 20, there appears to be a description that implies that expressive scores from one emotion were analyzed with receptive ability within the same emotion group.  Finally, the answer as to whether emotion expression is connected to emotion receiving appears to be some sort of average correlation (or a single measure of association that says generally emotion expressive ability is related to emotion receiving ability).   However, not all emotions are created equal.  What does the connection across the (6) different emotions categories look like?   Are some emotion categories connected better than others?  Is there an argument for some emotion categories being more relevant to other social constructs such as extraversion.   I would think imagine positive hedonic emotions far more likely to be related to extraversion than negative ones and that extraversion would be more emergent when the expression of happiness is coordinated with the reception of happiness than would be the case with other emotions such as anger and disgust.

Author Response

(The authors gave the same response as above.)

Round 2

Reviewer 1 Report

Comments and Suggestions for Authors

While revision was not as extensive as I would have expected, the authors responded to all comments and I believe the manuscript to be adequately improved for publication. 

Reviewer 2 Report

Comments and Suggestions for Authors

Your revision has more than adequately addressed my concerns.